# Ballou’s Ancestral Inbreeding Coefficient: Formulation and New Estimate with Higher Reliability

**DOI:** 10.3390/ani14131844

**Published:** 2024-06-21

**Authors:** Tetsuro Nomura

**Affiliations:** Department of Industrial Life Sciences, Faculty of Life Sciences, Kyoto Sangyo University, Kyoto 603-8555, Japan; nomurat@cc.kyoto-su.ac.jp

**Keywords:** ancestral inbreeding, inbreeding depression, purging, gene dropping, novel estimate, improvement of reliability

## Abstract

**Simple Summary:**

Deleterious recessive alleles causing inbreeding depression may be eliminated from populations through purifying selection facilitated by inbreeding. Providing evidence of this phenomenon (i.e., inbreeding purging) is of great interest for conservation biologists and animal breeders. Ballou’s ancestral inbreeding coefficient ($F_{{BAL}_{-}ANC}$) is one of the most widely used pedigree-based measurements to detect inbreeding purging, but the theoretical basis has not been fully established. In this report, the author gives a mathematical formulation of $F_{{BAL}_{-}ANC}$ and proposes a new method for estimation based on the obtained formula. A stochastic simulation suggests that the new method could reduce the variance of estimates, compared to the conventional gene-dropping simulation.

**Abstract:**

Inbreeding is unavoidable in small populations. However, the deleterious effects of inbreeding on fitness-related traits (inbreeding depression) may not be an inevitable phenomenon, since deleterious recessive alleles causing inbreeding depression might be purged from populations through inbreeding and selection. Inbreeding purging has been of great interest in conservation biology and animal breeding, because populations manifesting lower inbreeding depression could be created even with a small number of breeding animals, if inbreeding purging exists. To date, many studies intending to detect inbreeding purging in captive and domesticated animal populations have been carried out using pedigree analysis. Ballou’s ancestral inbreeding coefficient ($F_{{BAL}_{-}ANC}$) is one of the most widely used measurements to detect inbreeding purging, but the theoretical basis for $F_{{BAL}_{-}ANC}$ has not been fully established. In most of the published works, estimates from stochastic simulation (gene-dropping simulation) have been used. In this report, the author provides a mathematical basis for $F_{{BAL}_{-}ANC}$ and proposes a new estimate by hybridizing stochastic and deterministic computation processes. A stochastic simulation suggests that the proposed method could considerably reduce the variance of estimates, compared to ordinary gene-dropping simulation, in which whole gene transmissions in a pedigree are stochastically determined. The favorable property of the proposed method results from the bypass of a part of the stochastic process in the ordinary gene-dropping simulation. Using the proposed method, the reliability of the estimates of $F_{{BAL}_{-}ANC}$ could be remarkably enhanced. The relationship between $F_{{BAL}_{-}ANC}$ and other pedigree-based parameters is also discussed.

## 1. Introduction

Inbreeding is defined as a mating between relatives and is unavoidable in small populations [1]. Inbred individuals show a reduction in phenotypic performance, especially in fitness-related traits, which is a phenomenon known as inbreeding depression [1,2,3]. Inbreeding depression has been documented in various animal and plant species [1,4,5,6]. It has been considered that inbreeding depression is largely caused by partial dominance, i.e., the existence of partially deleterious recessive alleles, although over-dominance and epistasis may also play a role [3,7].

One may intuitively expect that the more inbred a population, the greater the severity of inbreeding depression. However, as more deleterious recessive alleles are exposed to inbreeding, they could be purged from the population by selection, consequently reducing the inbreeding depression [3,8,9]. This phenomenon has been referred to as inbreeding purging [3,9]. Providing evidence of inbreeding purging is not only theoretically but also practically of great interest since populations less affected by inbreeding depression could be established if inbreeding purging exists. To date, many studies intending to detect inbreeding purging have been carried out in captive wild [10,11,12], domesticated animal [13,14,15,16,17,18,19,20,21,22,23,24,25,26,27,28,29] and human populations [30], some of which suggests the existence of inbreeding purging [10,12,15,17,28,30]. However, it has been also shown that inbreeding purging is not a ubiquitous phenomenon [10,12,31].

In these studies, various pedigree-based measurements of purging opportunity induced through inbreeding in ancestors have been estimated (for a review, see [31]). One of the most widely used measurements is Ballou’s ancestral inbreeding coefficient ($F_{{BAL}_{-}ANC}$), which is defined as the cumulative proportion of an individual’s genome that has been previously exposed to inbreeding in its ancestors [10]. Irrespective of the wide use of $F_{{BAL}_{-}ANC}$ [12,13,14,15,16,17,18,19,20,21,23,24,25,26,28,30,31], the theoretical basis has not been fully established; in the pioneering work of Ballou [10], a recurrence equation to compute $F_{{BAL}_{-}ANC}$ from pedigree was given, but later it was shown that $F_{{BAL}_{-}ANC}$ computed from the equation overestimates the values obtained from stochastic simulation (gene-dropping simulation) [32]. To date, however, the exact computational procedure remains unknown. In all the published works, estimated values from gene-dropping simulation [12,13,14,15,16,17,18,19,20,21,23,24,25,26,28] or, as an approximation, values computed from Ballou’s equation have been used [12,30,31,33].

In the present report, I translate Ballou’s definition of $F_{{BAL}_{-}ANC}$ into a mathematical expression. Although the deterministic computation of $F_{{BAL}_{-}ANC}$ from the expression is limited to simple cases, from the expression a new stochastic method for estimating $F_{{BAL}_{-}ANC}$ with a higher reliability is proposed. The performance of the new method is evaluated by stochastic simulation. Finally, I discuss some theoretical aspects of $F_{{BAL}_{-}ANC}$ and the relation to other pedigree-based parameters to detect inbreeding purging.

## 2. Notation

Consider the pedigree of an individual *X* originated from *N_A_* founders (ancestors with unknown parents), each with unique alleles. Founders are described as *A*_1_, *A*_2_, …, $A_{N_{A}}$, and two alleles of founder *A_i_* are denoted as $a_{\left(i,1\right)}$ and $a_{\left(i,2\right)}$. We attach superscripts (0, *N*, 1) to $a_{\left(i,j\right)}$, according to its history of autozygosity:

$a_{\left(i,j\right)}^{0}$: the allele has experienced no autozygous state in the past,

$a_{\left(i,j\right)}^{N}$: the allele experienced an autozygous state for the first time in a given individual,

$a_{\left(i,j\right)}^{1}$: the allele has already experienced an autozygous state at least once.

Note that $a_{\left(i,j\right)}^{N}$ is a transient notation; it is immediately replaced by $a_{\left(i,j\right)}^{1}$ when transmitted to the next generation. The number of inbred ancestors in the pedigree is denoted as $N_{B}$, and the inbred ancestors are described as $B_{1}$, $B_{2}$,…, $B_{N_{B}}$.

The partial inbreeding coefficient [34] of individual *k* for founder allele $a_{\left(i,j\right)}$ is denoted as $F_{\left(i,j\right)k}$, which expresses the probability that individual *k* is autozygous for $a_{\left(i,j\right)}$, i.e., the genotype of *k* is $a_{\left(i,j\right)}a_{\left(i,j\right)}$. With the autozygous history notations, $F_{\left(i,j\right)k}$ is expressed by the sum of the three probabilities:$$F_{\left(i,j\right)k}=P_{k}\left(a_{\left(i,j\right)}^{1}a_{\left(i,j\right)}^{1}\right)+P_{k}\left(a_{\left(i,j\right)}^{1}a_{\left(i,j\right)}^{N}\right)+P_{k}\left(a_{\left(i,j\right)}^{N}a_{\left(i,j\right)}^{N}\right),$$
where $P_{k}(xy)$ is the probability that the individual *k* has the genotype xy. Due to symmetry in pedigree, $F_{(I,1)k}=F_{\left(i,2\right)k}$. Thus, the partial inbreeding coefficient ($F_{\left(i\right)k}$) of individual *k* for founder *i* is
$$F_{\left(i\right)k}=F_{\left(i,1\right)k}+F_{\left(i,2\right)k}=2F_{\left(i,1\right)k}=2F_{\left(i,2\right)k}.$$

By the definition of the partial inbreeding coefficient [34], Wright’s [35] classical inbreeding coefficient ($F_{k}$) of individual *k* is
$$F_{k}=\sum_{i=1}^{N_{A}}F_{\left(i\right)k}=\sum_{i=1}^{N_{A}}\sum_{j=1}^{2}F_{\left(i,j\right)k}=\sum_{i=1}^{N_{A}}\sum_{j=1}^{2}P_{k}\left(a_{\left(i,j\right)}^{1}a_{\left(i,j\right)}^{1}\right)+\sum_{i=1}^{N_{A}}\sum_{j=1}^{2}P_{k}\left(a_{\left(i,j\right)}^{1}a_{\left(i,j\right)}^{N}\right)+\sum_{i=1}^{N_{A}}\sum_{j=1}^{2}P_{k}\left(a_{\left(i,j\right)}^{N}a_{\left(i,j\right)}^{N}\right)$$

For the sake of simplicity, we write the above expression as
(1)$$F_{k}=P_{k}\left(a^{1}a^{1}\right)+P_{k}\left(a^{1}a^{N}\right)+P_{k}\left(a^{N}a^{N}\right).$$

## 3. Derivation of Expression

Denoting male and female parents of individual *X* as *S* and *D*, respectively, the recurrence equation given by Ballou [10] is
(2)$$F_{{BAL}_{-}ANC,X}=\frac{1}{2}\left[F_{{BAL}_{-}ANC,S}+\left(1-F_{{BAL}_{-}ANC,S}\right)F_{S}\right]+\frac{1}{2}\left[F_{{BAL}_{-}ANC,D}+\left(1-F_{{BAL}_{-}ANC,D}\right)F_{D}\right].$$

However, using a gene-dropping simulation, Suwanlee et al. [32] showed that Equation (2) overestimates $F_{{BAL}_{-}ANC,X}$. As a revised version of Equation (2), they proposed an equation [32]:(3)$$F_{{BAL}_{-}ANC,X}=\frac{1}{2}\left[F_{{BAL}_{-}ANC,S}+\left(1-F_{{BAL}_{-}ANC,S}|F_{S}\right)F_{S}\right]+\frac{1}{2}\left[F_{{BAL}_{-}ANC,D}+\left(1-F_{{BAL}_{-}ANC,D}|F_{D}\right)F_{D}\right],$$
where $F_{{BAL}_{-}ANC,S}|F_{S}$ is the proportion of genome of *S* that has been exposed to autozygosity at least once in the past, given that the genome is in an autozygous state in *S* (or equivalently, the conditional probability that an allele of *S* on an arbitrary locus has been exposed to autozygosity at least once in the past, given that the allele is in an autozygous state in *S*) and $F_{{BAL}_{-}ANC,D}|F_{D}$ is the similar proportion in *D*.

The term $\left(1-F_{{BAL}_{-}ANC,S}|F_{S}\right)F_{S}$ in Equation (3) implies the proportion of genome newly exposed to inbreeding in *S*, and $\left(1-F_{{BAL}_{-}ANC,S}|F_{D}\right)F_{D}$ is the similar proportion in *D*. Denoting these terms as $F_{{BAL}_{-}NEW,S}$ and $F_{{BAL}_{-}NEW,D}$, respectively, we have
(4)$$F_{{BAL}_{-}ANC,X}=\frac{1}{2}\left[F_{{BAL}_{-}ANC,S}+F_{{BAL}_{-}NEW,S}\right]+\frac{1}{2}\left[F_{{BAL}_{-}ANC,D}+F_{{BAL}_{-}NEW,D}\right].$$

Here we consider the simple pedigree shown in Figure 1. With Equation (4), $F_{{BAL}_{-}ANC}$ of individual *X* in the pedigree can be expanded to
$$F_{{BAL}_{-}ANC,X}={\left(\frac{1}{2}\right)}^{2}\left[F_{{BAL}_{-}ANC,K}+F_{{BAL}_{-}NEW,K}\right]+\left(\frac{1}{2}\right)\left[F_{{BAL}_{-}ANC,L}+F_{{BAL}_{-}NEW,L}\right]\;\;+{\left(\frac{1}{2}\right)}^{2}\left[F_{{BAL}_{-}ANC,O}+F_{{BAL}_{-}NEW,O}\right]+\left(\frac{1}{2}\right)\left[F_{{BAL}_{-}ANC,D}+F_{{BAL}_{-}NEW,D}\right]\;\;+\left(\frac{1}{2}\right)\left[F_{{BAL}_{-}ANC,S}+F_{{BAL}_{-}NEW,S}\right].$$

In this expression, ${\left(\frac{1}{2}\right)}^{n}$ is the genetic contribution [36] of ancestor (*K*, *L*, *O*, *S*, *D*) to *X*.

Analogously in any pedigree, we can expand $F_{{BAL}_{-}ANC}$ of an individual and can express it with $F_{{BAL}_{-}ANC}$ and $F_{{BAL}_{-}NEW}$ of all ancestors in the pedigree. But we should note that, by the definition of $F_{{BAL}_{-}ANC}$, an individual without inbred ancestors should have $F_{{BAL}_{-}ANC}=0$, and a non-inbred individual should have $F_{{BAL}_{-}NEW}=0$. This leads to a considerable simplification of the expanded form. In general, $F_{{BAL}_{-}ANC}$ of individual *X* can be expressed as
(5)$$F_{{BAL}_{-}ANC,X}=\sum_{k=1}^{N_{B}}{gc}_{\left(B_{k}\right)X}F_{{BAL}_{-}NEW,B_{k}},$$
where ${gc}_{\left(B_{k}\right)X}$ is the genetic contribution of inbred ancestor *B_k_* to *X*. This is an explicit expression of Ballou’s definition of the ancestral inbreeding coefficient. Note that $F_{{BAL}_{-}ANC,X}$ is expressed only with $F_{{BAL}_{-}NEW}$ of inbred ancestors and their genetic contributions.

As an application of Equation (5), consider the real pedigree shown in Figure 2, which is a part of the pedigree of a mare (individual *X*) in a captive population of Przewalski’s horse [37]. Such a pedigree will be typically found in the early history of a captive population expanded from a limited number of wild-caught founders. From Equation (5), $F_{{BAL}_{-}ANC}$ of individual *X* is expanded as
(6)$$F_{{BAL}_{-}ANC,X}=\sum_{k=1}^{3}{gc}_{\left(B_{k}\right)X}F_{{BAL}_{-}NEW,B_{k}}=\left(\frac{3}{4}\right)F_{{BAL}_{-}NEW,B_{1}}+\left(\frac{1}{4}\right)F_{{BAL}_{-}NEW,B_{2}}+\left(\frac{1}{2}\right)F_{{BAL}_{-}NEW,B_{3}}$$

To complete the computation of $F_{{BAL}_{-}ANC,X}$, we need to obtain values of $F_{{BAL}_{-}NEW,B_{k}}$ For the pedigree shown in Figure 2, it is trivial that $F_{{BAL}_{-}NEW,B_{1}}=F_{B_{1}}=0.125$ and $F_{{BAL}_{-}NEW,B_{2}}=F_{B_{2}}=0.125$, since these ancestors have no inbred ancestors. But the computation of $F_{{BAL}_{-}NEW,B_{3}}$ is complicated. Prior to the computation, we generally consider the allele-based expression of $F_{{BAL}_{-}NEW,B_{k}}$. Since $F_{{BAL}_{-}NEW,B_{k}}$ can be alternatively viewed as the expected frequency of the founder allele that is newly exposed to inbreeding in *B_k_*, $F_{{BAL}_{-}NEW,B_{k}}$ for a founder allele $a_{\left(i,j\right)}$ is written, with an analogy to the partial inbreeding coefficient $F_{\left(i,j\right)}$, as
$$F_{\left(i,j\right)BAL\_NEW,B_{k}}=P_{B_{k}}\left(a_{\left(i,j\right)}^{N}\right)=P_{B_{k}}\left(a_{\left(i,j\right)}^{N}a_{\left(i,j\right)}^{N}\right)+\frac{1}{2}P_{B_{k}}\left(a_{\left(i,j\right)}^{1}a_{\left(i,j\right)}^{N}\right).$$

Summing this expression over all founders and alleles within each founder and applying an analogy to Equation (1), we obtain the allele-based expression of $F_{{BAL}_{-}NEW,B_{k}}$ as
(7)$$F_{BAL\_NEW,B_{k}}=\sum_{i=1}^{N_{A}}\sum_{j=1}^{2}\left[P_{B_{k}}\left(a_{\left(i,j\right)}^{N}a_{\left(i,j\right)}^{N}\right)+\frac{1}{2}P_{B_{k}}\left(a_{\left(i,j\right)}^{1}a_{\left(i,j\right)}^{N}\right)\right]=P_{B_{k}}\left(a^{N}a^{N}\right)+\frac{1}{2}P_{B_{k}}\left(a^{1}a^{N}\right).$$

To apply Equation (7) to the computation of $F_{{BAL}_{-}NEW,B_{3}}$ in Figure 2, we introduce the nine condensed identity states (S_1_–S_9_ shown in Figure 3) [38] between the parents (*B*_1_ and *B*_2_) and their probabilities of occurrence (${∆}_{1}-{∆}_{9}$), that is, the condensed identity coefficients [38]. Four identity states (S_3_, S_5_, S_7_ and S_8_) are relevant to the computation of $F_{{BAL}_{-}NEW,B_{3}}$: from S_3_ and S_5_, a child with genotype $a^{1}a^{N}$ will be born with the probability 1/2, and from S_7_ and S_8_, a child with genotype $a^{N}a^{N}$ will be born with the probabilities 1/2 and 1/4, respectively (Figure 3). I obtained ${∆}_{3}=0.0625$, ${∆}_{5}=0.0625$, ${∆}_{7}=0.1094$ and ${∆}_{8}=0.4688$ from the ribd package [39] in R [40]. Thus, $P_{B_{3}}\left(a^{1}a^{N}\right)=\frac{1}{2}{∆}_{3}+\frac{1}{2}{∆}_{5}=0.0625$ and $P_{B_{3}}\left(a^{N}a^{N}\right)=\frac{1}{2}{∆}_{7}+\frac{1}{4}{∆}_{8}=0.1719$. Substituting these values into Equation (7) gives $F_{{BAL}_{-}NEW,B_{3}}=0.2031$. Finally, substituting the obtained values of $F_{{BAL}_{-}NEW,B_{k}}$ (*k* = 1, 2, 3) into Equation (6), we get $F_{{BAL}_{-}ANC,X}=0.2266$. The computational process is summarized in Table 1. The estimated $F_{{BAL}_{-}ANC,X}$ from a gene-dropping simulation with 10^6^ replicates using GRain (v2.2) [41,42] was 0.2263, while Ballou’s original Equation (2) gave an overestimated value as
$$F_{BAL\_ANC,X}=\left(\frac{1}{2}\right)\left[F_{BAL\_ANC,B_{1}}+\left(1-F_{BAL\_ANC,B_{1}}\right)F_{B_{1}}\right]+\left(\frac{1}{2}\right)\left[F_{BAL\_ANC,B_{3}}+\left(1-F_{BAL\_ANC,B_{3}}\right)F_{B_{3}}\right]\;\;=\left(\frac{1}{2}\right)\left[0+\left(1-0\right)\times 0.125\right]+\left(\frac{1}{2}\right)\left[0.125+\left(1-0.125\right)\times 0.25\right]=0.2344.$$

## 4. New Estimate and Its Performance

### 4.1. New Estimate of $F_{{BAL}_{-}ANC}$

If $F_{{BAL}_{-}NEW}$ of an individual with multiple ($N_{B}\geq 3$) inbred ancestors is considered, the deterministic computation is complicated since it requires the condensed identity coefficients among the multiple inbred ancestors. In theory, computation of the condensed identity coefficients among multiple individuals will be possible [43], but the possible number ($n_{s}$) of the condensed identity states exponentially increases as the number ($n_{d}$) of involved individuals increases; e.g., for $n_{d}$ = 3, $n_{s}$ = 66 and for $n_{d}$ = 4, $n_{s}$ = 712 [44]. In addition, we must trace the autozygous history of the alleles involved in each condensed identity state. These make the generalized computation of $F_{{BAL}_{-}NEW}$ intractable for a pedigree with multiple inbred ancestors. In fact, for the repeated full-sib mating, I found a compact set of recurrence equations which gives $F_{{BAL}_{-}NEW}$ and $F_{{BAL}_{-}ANC}$ at any generation. But unfortunately, the generalization seems to be hopeless.

As an alternative to the deterministic computation of $F_{{BAL}_{-}NEW}$, I propose the use of $F_{{BAL}_{-}NEW}$ estimated from a gene-dropping simulation. As shown later, estimates of $F_{{BAL}_{-}ANC}$ from this method have a favorable property. The simulation process is essentially same as the ordinary gene-dropping simulation [41,42]. In the simulation, alleles are flagged when they experience an autozygous state, and for an individual (necessarily, an inbred individual), newly flagged alleles ($a^{N}$ in our notation) are counted over replicates of the simulation. The total number of counts divided by the number of replicates of the simulation gives a stochastic estimate of $F_{{BAL}_{-}NEW}$ of the individual. Substituting the estimates of $F_{{BAL}_{-}NEW}$ for all inbred ancestors into Equation (5) gives an estimate of $F_{{BAL}_{-}ANC,X}$. This method is referred to as the ‘hybrid method’, in a sense that it consists of a stochastic process (the gene-dropping simulation) and a deterministic process (the computation of genetic contributions from the inbred ancestors).

As an illustrative example, we apply the hybrid method to the pedigree of Przewalski’s horse shown in Figure 2. Estimates of $F_{{BAL}_{-}NEW}$ for the inbred ancestors (*B*_1_, *B*_2_, *B*_3_) obtained from the gene-dropping simulation with 10^6^ replicates were 0.1250, 0.1243 and 0.2027, respectively. Weighting these estimates by the genetic contributions and summing them over all inbred ancestors (c.f., Equation (5)) gives $F_{{BAL}_{-}ANC,X}=0.2262$ as a hybrid estimate. The computational process is summarized in Table 1.

### 4.2. Performance of Hybrid Estimate

The performance of the hybrid estimate was evaluated with a more complicated pedigree (Figure 4), which is a part of the pedigree of the Spanish Habsburg dynasty [30,45]. Pedigree analysis with $F_{{BAL}_{-}ANC}$ has suggested that inbreeding purging had been acting in the early history of this dynasty [30]. I estimated $F_{{BAL}_{-}ANC}$ of King X (Charles II) from the hybrid method and evaluated the performance of the estimate by comparing with the estimate from the ordinary gene-dropping simulation. For both methods, 100 trials each with $n_{rep}$ (=5000, 10,000, 50,000 and 100,000) replicates were carried out. To fairly compare the two methods, I used the same simulation program originally coded in Fortran95. The estimated $F_{{BAL}_{-}ANC,X}$ from the gene-dropping simulation with 10^6^ replicates using GRain (v2.2) [41,42] was 0.2415.

The results of the simulation are summarized in Table 2, and for visualization, estimates from 100 trials of each method are plotted in Figure 5, for the case of $n_{rep}$ = 10,000. For a given number of replicates, the hybrid method reduced the variance of estimates to 40–60% of those from the ordinary gene-dropping simulation, implying that, by the use of the hybrid method, the reliability of the estimate could be enhanced.

## 5. Discussion

Equations (5) and (7) indicate that $F_{{BAL}_{-}ANC}$ of an individual is defined with ‘allele frequency’, but not directly with ‘autozygosity’ in the individual. Thus, as mentioned by several authors [18,22,33], a direct relationship between $F_{{BAL}_{-}ANC}$ and Wright’s classical inbreeding coefficient (*F*) is not expected. In many works [13,14,15,16,17,18,19,20], the correlation between $F_{{BAL}_{-}ANC}$ and *F* has been reported, ranging from 0.31 [20] to 0.95 [13]. The wide range of variation can be viewed as a reflection of the definition of $F_{{BAL}_{-}ANC}$.

In some cases, $F_{{BAL}_{-}ANC}$ and *F* will show quite different values; for example, if a population is subdivided into several isolated lines and mating between two lines occurs, the offspring will have *F* = 0, while $F_{{BAL}_{-}ANC}$ may show a positive value because of the accumulated $F_{{BAL}_{-}NEW}$ within the parental lines. Similarly, when a wild-caught animal with unknown parents is introduced into a captive population, a child of the introduced animal is expected to show *F* = 0 and $F_{{BAL}_{-}ANC}>0$ due to the accumulated $F_{{BAL}_{-}NEW}$ in the native parent of the child. In this context, $F_{{BAL}_{-}ANC}$ could be one criterion for selecting breeding animals. If other conditions are the same, an animal with a higher $F_{{BAL}_{-}ANC}$ should be preferred as a breeding animal to an animal with a lower $F_{{BAL}_{-}ANC}$, because the former is expected to have a smaller number of deleterious recessive alleles that may cause inbreeding depression in the descendants.

Kalinowski’s ancestral inbreeding coefficient ($F_{{KAL}_{-}ANC}$) is another measurement of purging opportunity induced by ancestral inbreeding [11]. This coefficient is based on a decomposition of *F* as $F=F_{{KAL}_{-}ANC}+F_{{KAL}_{-}NEW}$, where $F_{{KAL}_{-}NEW}$ is Kalinowski’s new inbreeding coefficient [11]. $F_{{KAL}_{-}ANC}$ is the probability that alleles are in an autozygous state in the individual while they have been in an autozygous state at least once in the past, and $F_{{KAL}_{-}NEW}$ is the probability that alleles are in an autozygous state for the first time in the individual [11,42]. In our notation,
$$F_{{KAL}_{-}ANC,X}=P_{X}\left(a^{1}a^{1}\right)+P_{X}\left(a^{1}a^{N}\right)$$
$$F_{{KAL}_{-}NEW,X}=P_{X}\left(a^{N}a^{N}\right).$$

Of course, as verified from Equation (1), $F_{X}=F_{{KAL}_{-}ANC,X}+F_{{KAL}_{-}NEW,X}$. Unlike $F_{{BAL}_{-}ANC}$, $F_{{KAL}_{-}ANC}$ has a direct relation to *F*. Thus, it is natural that a higher correlation has been found between $F_{{KAL}_{-}ANC}$ and *F* than between $F_{{BAL}_{-}ANC}$ and *F* in many works [14,15,16,17,18,19].

For the pedigree in Figure 2, $F_{{KAL}_{-}NEW,B_{3}}$ is computed from Figure 3 as
$$F_{{KAL}_{-}NEW,B_{3}}=P_{B_{3}}\left(a^{N}a^{N}\right)=\frac{1}{2}{∆}_{7}+\frac{1}{4}{∆}_{8}=0.1719.$$

Since $F_{B_{3}}=0.25$, we get
$$F_{{KAL}_{-}ANC,B_{3}}=F_{B_{3}}-F_{{KAL}_{-}NEW,B_{3}}=0.0781.$$

The corresponding estimates from GRain (v2.2) [41,42] with 10^6^ replicates are $F_{{KAL}_{-}NEW,B_{3}}=0.1711$ and $F_{{KAL}_{-}ANC,B_{3}}=0.0778$. However, the exact computation of $F_{{KAL}_{-}NEW}$ and $F_{{KAL}_{-}ANC}$ for an individual with multiple inbred ancestors will be intractable for the same reason as the difficulty in generalized computation of $F_{{BAL}_{-}ANC}$ and $F_{{BAL}_{-}NEW}$.

Gulsija and Crow [46] gave parameters to evaluate the potential reduction in an individual’s inbreeding load from pedigree data. Assuming that in the same path in a pedigree no two ancestors are autozygous for the same founder allele, they derived a parameter $O_{X}$ (opportunity for purging) to measure the opportunity for purging by the expected contribution of alleles from inbred ancestors to individual *X* [46]. The derived expression of $O_{X}$ is in our notation:$$O_{X}=\sum_{k=1}^{N_{B}}{gc}_{{(B}_{k})X}F_{B_{k}}$$

On the above assumption, it should be that $F_{{BAL}_{-}ANC,B_{k}}=0$ and $F_{{BAL}_{-}NEW,B_{k}}=F_{B_{k}}$. Thus, from Equation (5) we have
$$O_{X}=F_{{BAL}_{-}ANC,X}$$

For a complex pedigree involving several inbred ancestors in the same path, an inbred individual descending from inbred ancestors will be less likely to carry deleterious recessive alleles than when their ancestors have not been inbred [47]. To remove this remote ancestral effect from $O_{X}$, Gulsija and Crow [46] showed an equation. However, it seems to be too complex to implement in practice [47]. Note that Equation (5) contains only the terms with $F_{{BAL}_{-}NEW}$, implying that the remote ancestral effects are completely excluded from $F_{{BAL}_{-}ANC}$.

In the present study, the hybrid method was proposed for estimating $F_{{BAL}_{-}ANC}$ from pedigree data. Although the examined cases are limited, it was suggested that the method could enhance the reliability of the estimate, compared to the ordinary gene-dropping simulation [41,42]. This favorable property results from the bypass of a part of the stochastic process in the ordinary gene-dropping simulation; in the hybrid method, contributions of the estimated $F_{{BAL}_{-}NEW}$ to individual *X* are fully deterministically computed with a genetic contribution, whereas in the ordinary gene-dropping simulation, whole transmission of alleles from founders to individual *X* are subject to stochastic events (Mendelian segregations), which inevitably inflates sampling variance of the estimates.

Prior to the implementation of the hybrid method, finding inbred ancestors and computing their genetic contributions are required. The extra task can be easily overcome. Rapid algorithms are now available for computing the inbreeding coefficients [48,49], which allows us to find inbred ancestors in a large pedigree efficiently [50]. Genetic contributions can be systematically obtained by computing a lower triangle matrix $L=\left[l_{i,j}\right]$, where $l_{i,j}$ is the genetic contribution of *i* to *j* [51]. There is a simple algorithm for computing **L**, applicable to a large pedigree [52].

Although the exact computation of $F_{{BAL}_{-}ANC}$ with the obtained expression is limited to small and simple pedigrees, the expression deepens our understanding of $F_{{BAL}_{-}ANC}$. A useful outcome from the expression will be the hybrid method for estimating $F_{{BAL}_{-}ANC}$. The performance should be intensively evaluated under various scenarios in future studies.

## 6. Conclusions

In this article, the author provided a mathematical basis for $F_{{BAL}_{-}ANC}$ and proposed a new method for estimating $F_{{BAL}_{-}ANC}$_._ A stochastic simulation suggested that the new method could remarkably enhance the reliability of estimates, compared to the conventional gene-dropping method.

## Figures and Tables

**Figure 1 animals-14-01844-f001:**
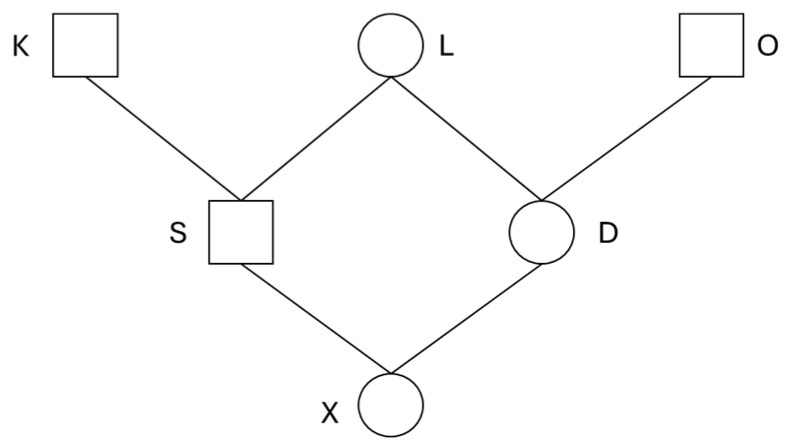
Simple pedigree used for explanation of computational procedure of *F_BAL_ANC_*.

**Figure 2 animals-14-01844-f002:**
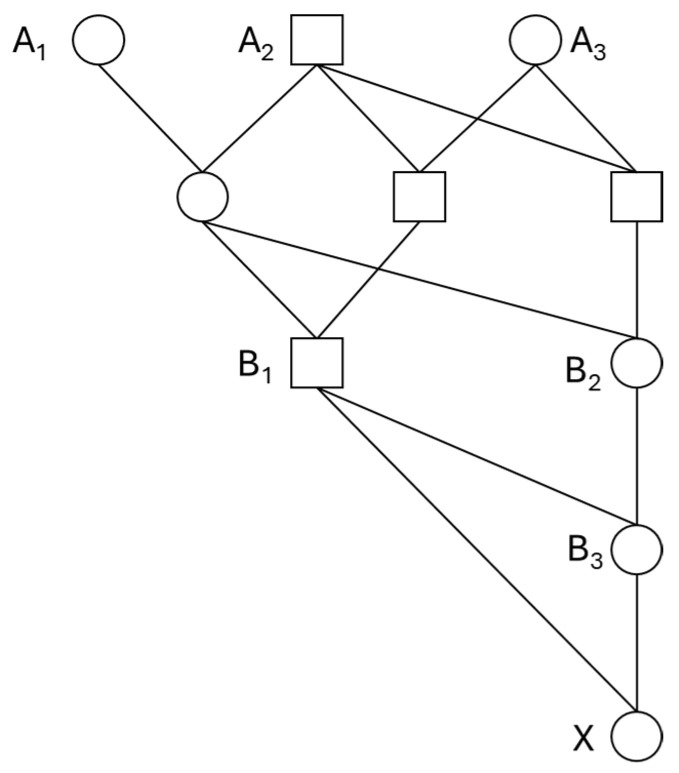
Pedigree of a mare (X) in captive population of Przewalski’s horse. *A_i_* (*i* = 1–3): founder, *B_k_* (*k* = 1–3): inbred ancestor.

**Figure 3 animals-14-01844-f003:**
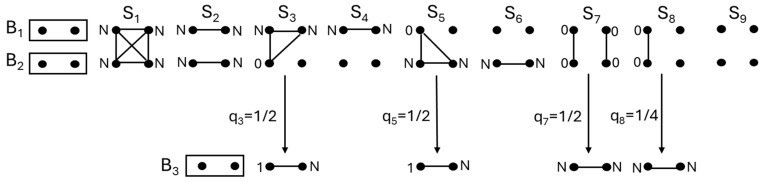
Nine condensed identity states (S_1_–S_9_) of individuals *B*_1_ and *B*_2_ in Figure 2. Dots linked with line are identical by descent, and dots not linked are not identical by descent. History of autozygosity for each allele is described with notations; 0: not exposed to autozygosity in the past, N: exposed to autozygosity for the first time in the individual, 1: exposed to autozygosity at least once in the past. For the four condensed identity states (S_3_, S_5_, S_7_ and S_8_) relevant to computation of $F_{{BAL}_{-}NEW,B3}$, the autozygous state of *B*_3_ born from each of the four parental identity states is shown with respect to the autozygous history (N, 1), together with the probability (q_i_) that the autozygous state occurs from each parental identity state.

**Figure 4 animals-14-01844-f004:**
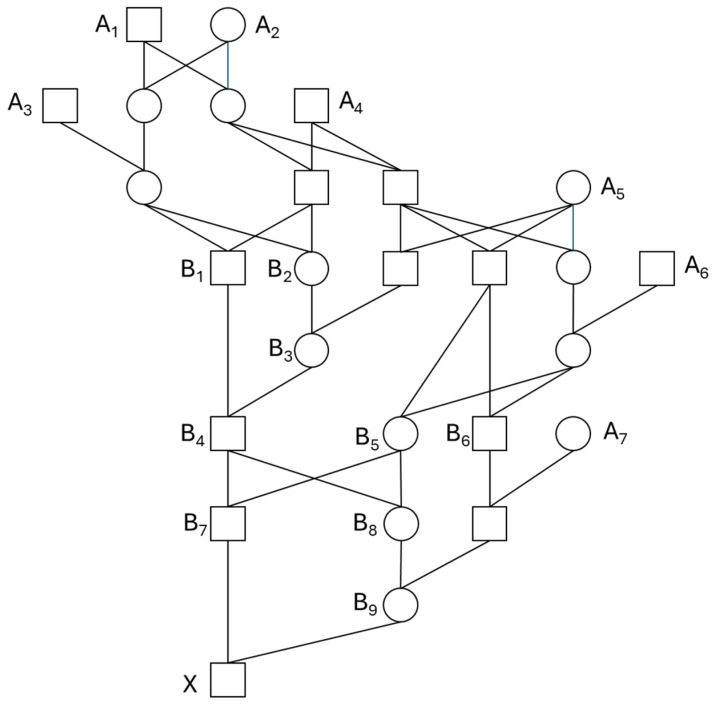
Part of pedigree of Spanish Habsburg dynasty. *A_i_* (*i* = 1–7): founder, *B_k_* (*k* = 1–9): inbred ancestor.

**Figure 5 animals-14-01844-f005:**
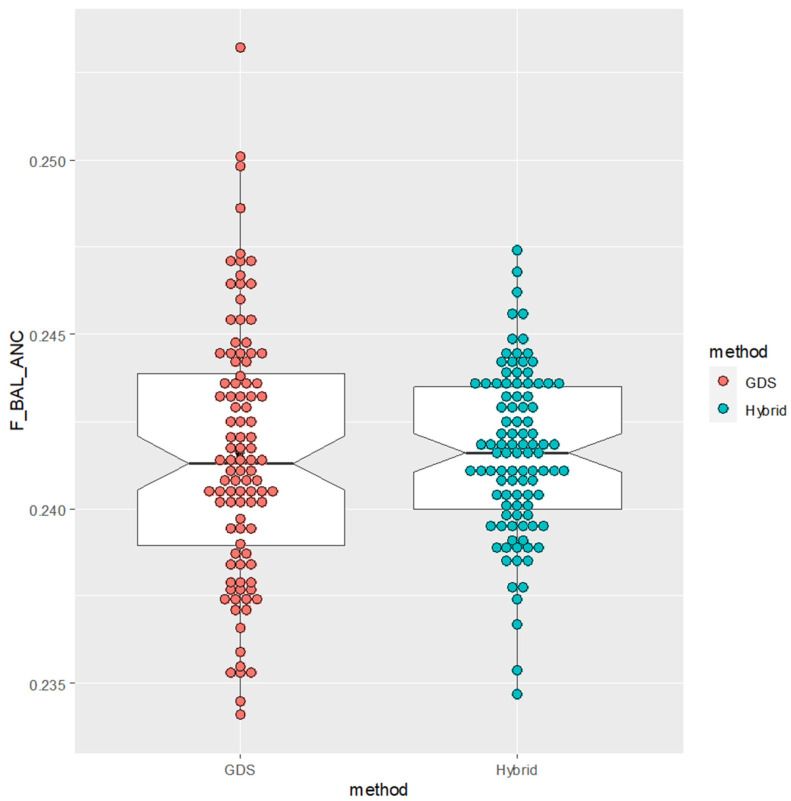
Plots of estimated *F_BAL_ANC_* of individual *X* in Figure 4, obtained from 100 trials of ordinary gene-dropping method (GDS) and hybrid method each with 10,000 replicates.

**Table 1 animals-14-01844-t001:** Computation of $F_{{BAL}_{-}ANC}$ of mare *X* in pedigree of Przewalski’s horse shown in Figure 2, from exact computation and hybrid method. *gc*: genetic contribution.

		Exact Computation		Hybrid Method	
Inbred Ancestor	*gc* to *X*	*F_BAL_NEW_*	gc× *F_BAL_NEW_*	*F_BAL_NEW_*	gc× *F_BAL_NEW_*
*B* _1_	0.75	0.125	0.0938	0.1250	0.0938
*B* _2_	0.25	0.125	0.0313	0.1243	0.0311
*B* _3_	0.5	0.2031	0.1016	0.2027	0.1014
		*F_BAL_ANC,X_* = 0.2266		*F_BAL_ANC,X_* = 0.2262	

**Table 2 animals-14-01844-t002:** Summary statistics of estimated *F_BAL_ANC_* of individual *X* in Figure 4, obtained from 100 trials each with *n_rep_* replicates. GDS: ordinary gene-dropping simulation, Hybrid: hybrid method. Figure in parentheses is fraction (%) of variance of estimates from hybrid method to that from GDS.

	$n_{rep}=$ 5000		$n_{rep}=$ 10,000		$n_{rep}=$ 50,000		$n_{rep}=$ 100,000	
	GDS	Hybrid	GDS	Hybrid	GDS	Hybrid	GDS	Hybrid
Average	0.2424	0.2421	0.2416	0.2416	0.2420	0.2417	0.2417	0.2415
Median	0.2424	0.2425	0.2413	0.2416	0.2421	0.2416	0.2418	0.2414
Variance × 10^6^	2.2067	1.2521 (56.7%)	1.3429	0.5534 (41.2%)	0.2345	0.1260 (54.2%)	0.1096	0.0649 (59.3%)

## Data Availability

All the numerical results presented in this article are available by reproducing with the R script and Fortran code given as Appendix A.

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
