# Peer review of "Ballou’s Ancestral Inbreeding Coefficient: Formulation and New Estimate with Higher Reliability"

_animals, 2024, doi:10.3390/ani14131844_

Round 1

Reviewer 1 Report

Comments and Suggestions for Authors

Animals-3042403-peer-review-report-v1.

Preamble

The author provides a general overview of inbreeding purging and introduces a new method for estimating it using the FBAL-ANC coefficient. This method combines stochastic and deterministic processes, significantly reducing estimate variance compared to traditional methods. The paper also discusses the relationship between 𝐹𝐡𝐴𝐿𝐴𝑁𝐢 and other pedigree-based parameters. The author used relevant references while writing the paper. However, the following suggestions could further improve the paper.

Abstract

a.     The abstract needs to provide adequate background information on the worth of inbreeding purging in conservation biology and animal breeding, so readers may not fully grasp the implications of the research.

b.     The abstract could include a statement about the research's implications, even if specific results or findings are not yet available. This will help readers understand the potential value of the intended method and its implications for understanding inbreeding purging.

c.      The abstract briefly mentions that the author provides a mathematical basis for FBAL-ANC and proposes a new estimate. However, it could be more explicit in highlighting the novelty of this method, which combines stochastic and deterministic processes, and its potential to significantly reduce estimate variance compared to traditional methods. This will help readers understand the unique contribution of your research.

d.    Line 17:  Add the articlethebefore the word deleterious.

e.     Line 21:  The worddatashould be date?

f.      Line 25: Add a hyphen between gene and dropping (gene-dropping).

g.     Change the prepositionoftoforto make the sentence grammatically correct.

h.     Lines 26/27: For the sake of clarity, rewrite this sentence. Suggestion: In this report, the author provides a mathematical basis for FBAL-ANC and proposes a new estimate by hybridizing stochastic and deterministic computation processes. 

i.       Addressing these shortcomings could improve the clarity and impact of the abstract, making it more informative and compelling to readers.

j.       Line 40: Add a hyphen between the wordsoveranddominance’.

k.     Line 43: Delete the phrasemore and(However, as more deleterious…).

l.       Lines 48-52: Rephrase this sentence and break it into two sentences for your readers to understand better. Suggestion: To date, many studies intending to detect inbreeding purging have been carried out in captive wild [10-12], domesticated animals [13-29] and human populations [30], some of which suggest the existence of inbreeding purging [10,12,15,17,28,30]. However, it has also been shown that inbreeding purging is not a ubiquitous phenomenon [10,12,31].

m.   Line 62:  Add the articlethebefore the word exact.  

n.     Line 66:  Add the articlethebefore the word deterministic.

2. Notation

a.     Line 73:  Add the articlethebefore the word pedigree.

b.     Line 77:  Add the articlethebefore the word past.

c.      Line 78:  Add the articlethebefore the word first.

d.    Line 79: This sentence requires recasting. Suggestion: The allele has already experienced an autozygous state at least once.

e.     Line 86:  Add the articlethebefore the word sum.

Discussion

a.     Line 279:  Add the articlethebefore the word correlation.

b.     Line 289:  Add the articlethebefore the word same.

c.      Line 297:  Add the articlethebefore the word past.

d.    Line 320/321:  Add the article ‘an’ before the word individual’s.

e.     Line 337: The conjunctionButis misused. Suggestion: Use the wordHoweverinstead.

f.      Lines 342-344: This sentence needs a reference.

g.     Line 344:  Add the articlethebefore the wordsbypassandstochastic’.

h.     Lines 345/347: Can a hyphen be added between the words gene and dropping? Check the whole paper; it may appear in many other areas. Adopt whichever one is correct.

i.       Line 351:  Add the articlethebefore the word implementation.

j.       Lines 352-354: This sentence could be rephrased to improve clarity. Suggestion: Rapid algorithms are now available for computing the inbreeding coefficients [49,50], which allows us to find inbred ancestors in a large pedigree efficiently.

k.     The discussion ended rather abruptly with no conclusion. I suggest you state any limitations confronted throughout the study that may have impacted the analysis and interpretation of the findings generated from this study.

l.       Briefly mention future research directions or next steps.

Comments on the Quality of English Language

They are included in the main report.

Author Response

Thank you for your helpful comments on my manuscript. According to your comments, I have made the following corrections:

Abstract

  1. Information on the worth of inbreeding purging in conservation biology and animal breeding has been given (Lines 21-22).
  2. Sentences briefly stating the potential value of the proposed method has been added (Lines 29-34).
  3. To stress the novelty and uniqueness of the proposed method, a sentence has been added (Lines 29-34).

d-n. All the suggestions have been reflected in the revised manuscript.

Notation

a-e. All the suggestions have been reflected in the revised manuscript.

Discussion

a-j. All the suggestions have been reflected in the revised manuscript.

K-l. At the end of Discussion, a new paragraph is added, in which the limitation and implication of the proposed method and the future direction of this study are briefly stated (Lines 358-362).

Reviewer 2 Report

Comments and Suggestions for Authors

The author exxamined the calculation possibilities of the anvestral inbreeding coefficient of Ballou (1997). The topic has high importance as this inbreeding coefficient can be used in order to evaluate the possible purging of the harmful genes in a given population. The topic also falls to the scope of the journal. Reading the manuscript I have to conclude that it was very interesting and nicely peresented. The cited literaure was adequate. The numarical examples were clear. I have to note however that in general the number of repetition in Grain is 1.000.000 therefore I am not sure if the results with smaller repetitions in Tabéle 2. has any relevance. Besides, the author should clearly explain in what sense the variance of the successive drops are important? We only use the mean of these estimates anyhow. I also would delete lines 351-357 as it is not directly related to the present work.

Author Response

  1. I would like to remain the paragraph (Lines X351-357), because it is necessary for the practical application of the proposed method. For the convenience of readers who try the proposed method, I also gave the R script for finding inbred ancestors and computing their genetic contributions as Supplementary materials.
  2. Both the proposed method and ordinary gene-dropping simulation will give unbiased estimates. However, under a fixed number of replications, there will be a difference of reliability between the two methods. Variance of estimates among trials (100 trials in this study) is an indicator of reliability. Although researchers may empirically evaluate the reliability of estimate by trying several runs in their preliminary analysis, they usually do not know the exact reliability. Since the purpose of the present simulation is the exact comparison of the two methods, I consider that the variance of estimates is essentially important. The smaller number of replicates (<1,000,000) are also considered to be necessary to grasp the performance of the proposed method.

Reviewer 3 Report

Comments and Suggestions for Authors

Ballou’s Ancestral Inbreeding Coefficient: Formulation and 2 New Estimate with Higher Reliability

Keywords: Ballou’s ancestral inbreeding coefficient; inbreeding purging; inbreeding depression; 31 formulation; new estimate

There are key words in the title. These should only appear in one place.

In line 90, inside the parenthesis it is empty:”… where π‘ƒπ‘˜( ) is the probability…!

The work is good, but very specific.

Authors must present supporting material (Supplementary) files that allow anyone who wants to do something similar to see and follow the step-by-step instructions. As well as listing the R packages used in the material and method.

Where is it: Fast Algorithms are now available to calculate inbreeding coefficients [49,50], which allow you to efficiently find inbred ancestors in a large pedigree. Need show in supplementary material. Just like spreadsheets and scripts in R program

The paper does not have a conclusion. This is fundamental

Author Response

  1. According to the comment, key words have been changed.
  2. The expression Pk() has been modified as Pk(xy) (Line 95).
  3. According to the suggestion, the following supplementary materials are given; R script for finding inbred ancestors and computing their genetic contributions. Fortran code used in the simulated comparison of the proposed method and ordinary gene-dropping method.
  4. At the end of Discussion, a new paragraph briefly stating the conclusion and the future direction of this study has been added (Lines 358-362).

Round 2

Reviewer 1 Report

Comments and Suggestions for Authors

So far so good.